# Development Pattern of Medical Device Technology and Regulatory Evolution of Cataract Treatment

**DOI:** 10.3390/healthcare11040453

**Published:** 2023-02-04

**Authors:** Heejung Kim, Harry Jeong, Kwangsoo Shin

**Affiliations:** 1Department of Biomedical Convergence, College of Medicine, Chungbuk National University, Chungdae-ro 1, Seowon-gu, Cheongju 28644, Chungbuk, Republic of Korea; 2Central Research Institute, Dr. Chung’s Food Co., Ltd., Cheongju 28446, Republic of Korea; 3Graduate School of Public Health and Healthcare Management, The Catholic University of Korea, Banpo-daero 222, Secho-gu, Seoul 06591, Republic of Korea; 4Catholic Institute for Public Health and Healthcare Management, The Catholic University of Korea, Banpo-daero 222, Secho-gu, Seoul 06591, Republic of Korea

**Keywords:** intraocular lens, medical device, healthcare technology innovation, regulatory evolution

## Abstract

To prevent regulation from becoming an obstacle to healthcare technological innovation, regulation should evolve as new healthcare technologies are developed. Although regulation is closely related to healthcare technology development, there are few studies that view healthcare technological advances from the multi-layered perspective of papers, patents, and clinical research and link this with regulatory evolution. Therefore, this study tried to develop a new method from a multi-layer perspective and draw regulatory implications based on it. This study applied this method to intraocular lens (IOLs) for cataract treatment and detected four major healthcare technologies and two recent healthcare technologies. Moreover, it discussed how current regulations evaluate these technologies. The findings provide implications for healthcare technological advances and the evolutionary direction of regulation through the example of IOLs for cataract treatment. This study contributes to the development of theoretical methods for co-evolution with regulations based on healthcare technology innovation.

## 1. Introduction

Emerging technology is not only economically beneficial, but can also create new industries or change existing industries, which can have great economic and industrial impacts on our society [1]. Thus, managers, researchers, and policymakers are trying to understand new technologies in their fields [2]. In particular, governments have established policies to detect innovative technologies, allocate budgets, prepare roadmaps for implementation, and support the rapid development of products incorporating innovative technologies. In the healthcare field, there are cases where the current regulations are inappropriate for application in relation to innovative healthcare products; thus, regulators may need to revise their regulations in advance [3]. Therefore, regulatory authorities should preemptively develop guidelines for verifying the safety and effectiveness of healthcare products to which innovative technologies are applied, provide them to developers, and if necessary, revise the relevant regulations.

Among healthcare products, medical devices occupy a lower proportion in healthcare than pharmaceuticals, although they are important for understanding healthcare technology because of their high relevance to clinical practice [4]. In particular, among medical devices, IOLs have a long development history and are widely used still today, and thus they offer representativeness when it comes to observing the development pattern of medical device technology [5]. Cataract, a disease associated with IOL implantation, was predicted to cause blindness in 13.5 million people worldwide by 2020 [6]. In 2017, 141.2 per 1000 adults in the United States had a visual impairment due to cataract [7]. Thus, in the United States, 3.33 million IOL implantations were performed in 2018 [8], and the cost of cataract surgery ranked 5th among all surgical costs in 2017 [9]. Additionally, IOLs are in line with the current medical device development trend for improving the quality of life. New products are developed in response to the needs of patients, who want to comfortably view various distances without glasses, such as outdoor activities, smartphone use, and reading.

Although the analysis of medical device development patterns is necessary to understand healthcare technology, there are few studies concerning medical device development patterns. Moreover, there are few studies on development patterns that include clinical trials, which are necessary for verifying the effectiveness and safety of medical devices. In other fields, such as chemistry and biotechnology, there are many studies that analyze new technologies by comparing patents and the literature [10,11,12,13,14,15,16,17]. However, research on medical device technology development patterns was conducted only on wearable devices and smart health monitoring systems [18,19]. They are relatively recent technologies and have limitations in terms of generalizing them to medical devices. Additionally, an important consideration in the medical device development pattern research is that medical devices must be approved by the regulatory authorities. However, there have been few studies that analyzed technology development patterns that included the clinical trial data required for medical device approval. Therefore, in this study, we proposed a new multi-layer analysis method involving patents, papers, and clinical trials to analyze medical device development patterns to understand healthcare technologies.

This study will contribute to the co-evolution of medical device technology and regulation. Current regulatory authorities are trying to enhance national competitiveness by not only approving healthcare products, but also detecting and supporting innovative healthcare products. This study indicated that even regulatory authorities can find innovative healthcare products based on objective data. It can also be used for regulation reviews and guideline development for safety and effectiveness evaluation. Furthermore, a multi-layer analysis using patents, papers, and clinical trial data can help governments, developers, and managers with decision making by providing information on technology development stages, such as research in academia, company-led development, and clinical trials [20]. This result will be able to provide useful information for the healthcare industry by increasing the understanding of the technology development patterns of healthcare products.

The remainder of this paper is structured as follows. In Section 2, we present the methods and text-mining methods applied to compare and analyze data from patents, papers, and clinical trials. In Section 3, we derive the major technologies applied to IOLs through a patent word cloud and topic modeling, and we bridge the derived technology with papers and clinical trials. Thus, we represent them as data lanes. Additionally, because our research goal is to develop a method that can be used for horizon scanning to derive new technologies in the field of medical devices, we prove that our method can discover new technologies that have recently appeared in IOL technology development. Finally, in Section 4, based on these results, we outline improvements that should be applied by regulators.

## 2. Methods

We evaluated the technological development patterns and newly emerged technologies of IOLs using a text-based analysis method using three types of data: patents, papers, and clinical trials. We used the methodologies of Niemann et al. [21], Block et al. [22], and Wustmans et al. [23] to compare the patents, papers, and clinical trials. In other words, if the words used in each document were similar, it was assumed that the content was also similar. Using this method, Niemann et al. [21] analyzed the similarity between patents, Block et al. [22] analyzed the similarity between patents and papers, and Wustmans et al. [23] analyzed the similarity between patents and trend data. Our research method is illustrated in Figure 1.

### 2.1. Patent Analysis

First, we tried to grasp the technology of IOLs from patents among patents, papers, and clinical trials for the following reasons. Patents contain much information about technology. Indeed, it is claimed that 80% of technology information is contained in patents [24]. Additionally, the number of patent applications is gradually increasing, indicating the importance of patents as a source of information [24]. In particular, companies use patents to protect intellectual property rights associated with information about high-value technologies [25]. Finally, the fact that patents are publicly available and the number of them is large also makes them suitable as a source for technology development research [26].

We searched for US patents among the patents provided by the Korean Intellectual Property Office (KIPO) using the Korea Intellectual Property Rights Information Service (KIPRIS) of the KIPO. We used US patents because the US is the largest of the global IOL markets and three of the five companies (Alcon, Inc., Johnson and Johnson Vision Care, and Bausch + Lomb) that lead the global market are US companies [27]. We used A61F2/16, an International Patent Classification corresponding to the IOL, as a search term. However, because patents on the IOL apparatus or IOL insertion method were searched, we additionally selected patents with “intraocular lens” in the title to reduce the false positives. The search period was from 1980 to 2021. This is because, although we tried to collect as much data as possible on IOLs, we could not obtain patents for IOLs prior to 1980 through the KIPRIS.

The analysis of a patent can be divided into four stages. The first step was data extraction and word refinement. Among the IOL patents defined above, words with parts of speech (POS) as nouns were extracted from the abstract. A list of thesauruses, defined words, and exceptions was made in the process of determining words with similar meanings, words with new meanings by combining several words, and words unnecessary for analysis. For example, both “haptic” and “haptics” are words that mean the supporting components of an IOL, although they can be written in different forms depending on the number; therefore, we needed to define them as synonyms. Additionally, posterior capsular opacification (PCO) is one of the major complications that occur after IOL implantation, but when extracted as “posterior”, “capsular”, and “opacification”, the original meaning cannot be derived. Therefore, we created a list of defined words for cases in which several words are combined to represent a new meaning. Moreover, words that are commonly included in almost all patents, such as “the eye”, or whose meaning is not derived from individual words, such as “section” and “portion”, were unnecessary in the analysis. Therefore, we excluded these words from the analysis by setting them as exceptions.

The second step was to construct a two-mode network composed of documents for extracting keywords and topics from the patents and then to analyze the properties of the degree, closeness, betweenness, and eigenvector centrality between each word (node) for the semantic network analysis. Degree centrality refers to the sum of other nodes connected to a specific node, and it is an indicator of the importance of the node through a direct connection relationship. Closeness centrality measures centrality through the minimum distance between other nodes connected to a specific node, betweenness centrality measures how much a broker plays a role between nodes, and eigenvector centrality is an index that measures the importance of a node [28].

The third step involved the derivation of topics related to IOLs shown in the patents and keywords constituting each topic through topic modeling. The Latent Dirichlet Allocation (LDA) algorithm was used in the topic modeling. The LDA is a probabilistic modeling method that finds topics through the words in each patent and determines the words that are placed in the topic. Since the LDA is an unsupervised generative model without prior information about the topics and words, we can easily find all the words related to the topic. However, it is not suitable for a small amount of data from a normally distributed environment [29].

In the last stage of the patent analysis, we analyzed the frequency of the appearance of the refined keywords and visualized them through a word cloud. In this way, we were able to derive the main topics of the IOL technology and topic words. Additionally, to derive the recent technology, it was necessary to observe the change in technology over time; therefore, we analyzed the patents by dividing them into five-year units. Although we set the search period from 1980 to 2021, since the US IOL patents matching our searching conditions were searched since 1986, we analyzed them by dividing them into time periods of 1986–1990, 1991–1995, 1996–2000, 2001–2005, 2006–2010, 2011–2015, and 2016–2021. However, we were unable to obtain the analysis results for 2001–2005 because no IOL-related patents were searched during this period. Moreover, patent data published in 2021 were combined with data from 2016–2020.

### 2.2. Multilayer Analysis

After completing the patent analysis, we analyzed how the major and recent IOL technologies derived from the patents appeared in the papers and clinical trials. The main technologies of IOLs shown in the patents were “haptic”, “material”, “surface”, and “accommodation” (Figure 2). The recent technologies were “depth of focus” and “injector”. Therefore, we extracted papers containing the keywords for these four topics in the title from the Scopus database and clinical trials containing the keywords in a brief summary from the ClinicalTrials database. Subsequently, we created a data lane by combining the patents, papers, and clinical trials.

At first, we searched for papers with “intraocular lens” in the title using the Scopus database (DB). The Scopus database is produced by Elsevier Co., and it provides information on more than 87 million documents, more than 1.8 billion cited references, and more than 7000 publishers [30]. Thus, it is suitable for the retrieval of scientific papers. From our search, we found more articles on clinical trials of IOLs than articles on technology. As we were going to use the ClinicalTrials database rather than papers on clinical trials, we needed to exclude the literature on clinical trials from the paper search results. Therefore, we excluded papers containing “clinic*” in the title, abstract, or keywords from the search results. Additionally, because we were going to analyze how the major IOL technologies appearing in the patents appeared in the papers and clinical trials, we searched for papers containing the major technologies derived from the patents in the title. We searched for literature published between 1986 and 2021. This is because IOL patents have been obtained since 1986. The comparison of the three databases was important in this study, as it was necessary to analyze patents, papers, and clinical trials during the same period.

Second, we analyzed the clinical trials using the ClinicalTrials database, which is a database of clinical studies maintained by the National Library of Medicine of the National Institutes of Health in the US. The database was developed to increase public access to clinical trials, and it now provides information on more than 400,000 studies in 220 countries as well as 50 states in the US [31]. We extracted clinical trials that included “intraocular lens” in “condition or disease” from the database and searched for clinical trials that included key technologies derived from the patents and papers. When searching the ClinicalTrials database, we set the search period from 1986 to 2021 in the same manner. However, this database has only required the registration of clinical trials and results information since 2007 [31]. Therefore, data from before 2007 may not reflect all the clinical trials.

### 2.3. Focus Group Interview

We selected a focus group of experts and verified the results of our analysis. We wanted to verify the validity of our analysis method and results, and for this we conducted focus group interviews. Five experts were selected for the study. First, we included an IOL developer because we wanted to include experts in IOL R&D. The expert was also in charge of developing national standards for the field of ophthalmology in Korea. Additionally, we included clinicians who selected IOLs and performed cataract surgery in the focus group. We also included a reviewer from the regulatory authorities in the Korean government and an expert who tested the performance of IOLs in the nationally accredited testing institute in the focus group. Finally, to prevent the focus group from being composed of only medical experts, we verified our analysis results by including experts with extensive experience in technology development in the non-medical device field.

Based on our results, we prepared the interview questions in advance before meeting with each expert. The questions included the research methods and research results, and they covered the following topics:-Validity of the research method;-Suitability of IOL technologies derived through the word cloud and topic modeling;-Interpretation validity of the relationships of the words in the topic modeling;-Validity of the explanations for the changes in technologies over time;-Suitability of the explanations for bridging the patents, papers, and clinical trials;-Suitability of the conclusions drawn by bridging the patents, papers, and clinical trials;-Other expert opinions from the analysis results.

We asked the same questions to all five experts, and our questions are listed in Table 1.

## 3. Results

First, the main IOL technologies shown in the patents were visualized as a word cloud and topic modeling over time (Figure 2, Figure 3 and Figure 4). The word cloud analysis results can identify the words frequently mentioned in the patents. Generally, the higher the frequency of appearance, the larger the font size of the text. As a result of the analysis, four major themes were derived: haptic, material, surface, and accommodation. The topic modeling results provide us with a slightly deeper insight into the four derived topics, as they provide words related to the technologies. Second, to show that we can discover the latest IOL technological development trends using this method, we analyzed the technology development trends over the last six years.

When performing topic modeling, the number of topics to be extracted must be specified. In previous studies, the analysis was performed by designating 5–10 topics, with the number of topics being determined based on the interpretability after the analysis by repeatedly changing the number of topics [32]. This study was analyzed in the same way, and the optimal number of topics was seven.

The following can be derived for each of the four major IOL technologies from the topic modeling. First, haptic, which is closely related to IOL stability, has been developed in various forms to increase IOL stability, and it has recently become the accommodation core of AIOLs. From the topic modeling, haptic was highly correlated with the posterior chamber, capsular bag, and fixation (1986–1990, topic 1). As haptic is highly related with IOL stability, it is considered to be highly related to the success or failure of IOL implantation surgery [33]. In particular, haptic is known to be highly related to the intraocular stability of products with enhanced optical functions, such as multifocal (MIOLs) and toric IOLs [33].

From the topic modeling, plate, ribbon, and haptic were found to be highly related (2011–2015, topic 5), which can be interpreted as various types of haptics being developed to increase IOL stability. The literature also confirmed that C-shaped; modified C-loop 2 haptics; three, four, or six haptics with improved capsular bag stability; and plate-haptics suitable for toric IOL have been developed depending on the shape and function of the IOL [33].

Additionally, AIOLs and haptics showed high correlation (2011–2015, topic 5) because haptics control the visual acuity of AIOLs. In AIOLs, the contraction of the ciliary muscle compresses the haptic and moves the optic forward; therefore, the compressed haptic is an important part of the accommodation technique [34].

In terms of the second technology, IOL material occupied a major part of the IOL development because it should have suitable properties to replace the human crystalline lens. From our analysis, the material was not only a major word during all the analysis periods, but also showed a close relationship with polymer, mixture, and composition (1991–1995, topic 5). This indicated that there has been technological development wherein various monomers are polymerized with different compositions to develop an IOL material with suitable characteristics. The lens material should be biocompatible, capable of optical performance, stable when implanted in the eye, and thin and flexible for IOL implantation [35]. As there is a trend of developing IOL materials with these characteristics, the material appears to be one of the main IOL technologies. In the literature, we found technological developments, such as the copolymerization of silicone with other monomers to obtain a material with a suitable refractive index or adjusting the IOL thickness [36].

IOL material technological development was used to reduce the side effects of cataract surgery, such as PCO. From the analysis, silicon appears in the word cloud. A previous study showed that silicone had a lower rate of PCO than other materials owing to its low cell deposition, which can be regarded as a technology development aimed at lowering the PCO rate [36]. Additionally, PMMA has good biostability and a low inflammatory response as a material used in early generations [36]. However, the recovery period is prolonged because an incision as large as the IOL size has to be made; therefore, it has been gradually replaced with flexible material [36]. In our results, this material change also appeared in the topic modeling (1986–1990, topic 7).

The following conclusions can be drawn for the third technology (surface). In the early days, the surface was mainly related to the materials; thus, there was an attempt to increase the biocompatibility of the eye and the IOL. However, over time, a multifocal, toric IOL technology developed to increase the independence of the patient’s glasses.

The topic modeling results showed a high relationship between the surface and material at the beginning (1986–1990, topic 7; 1996–2000, topic 1). Interactions between ocular tissue and IOLs can lead to complications, such as postoperative inflammation, cell and pigment deposition on the IOL, posterior synechiae, and capsule opacification [37]. To reduce these side effects, materials and surface processing with improved biocompatibility were developed. For example, studies on the cytologic features and blood–aqueous barrier related to IOL materials [38], as well as inflammation and capsule opacification according to the surface treatment, were conducted [37].

However, the surface gradually became more relevant to the visual acuity, which was related to the IOL technological development to provide two or more fixed optical powers and astigmatism corrections. Several IOLs use refractive or diffractive approaches, or a combination of both, to provide near, intermediate, and distant vision [39]. The high correlation between the surface and optical characteristics is related to the technological development of IOLs with aspheric surfaces and IOLs for astigmatism. To reduce the natural spherical aberration of the eye and provide more accurate vision than spherical IOLs, technology was developed to allow for the occurrence of positive spherical aberration by applying asphericity to the first or second surface of aspheric IOLs [40]. Approximately 40% of patients who need cataract surgery have corneal astigmatism that needs correction; thus, there has been toric IOL technological development for the complete glass independence of patients [41].

The following can be observed for the fourth technology, namely accommodation. Accommodation technology emerged owing to the desire to overcome presbyopia and improve the quality of life of patients. Although it appeared relatively later, it became a major IOL technology comparable to the other technologies. Presbyopia, a natural aging phenomenon, is caused by a decrease in lens elasticity, an increase in the equatorial diameter, the loss of Bruch’s membrane, and a decrease in ciliary muscle contractility [34]. Vision loss caused by cataracts could almost be overcome with IOL implantation. However, the only part that has not been recovered by surgery is “accommodation” [42], and to overcome this, AIOLs were developed [43].

We can compare our analysis results with the time when the AIOL was developed. We found that accommodating IOLs appeared from 2000 in the papers, in 2006–2010 in the patents, and in 2010 in the clinical trials. The Crystalens AT-45 (Eyeonics, Inc.), the first AIOL in the US, was approved in 2004 to provide near, intermediate, and distant vision without spectacles [44]. However, this lens was the first approved IOL in the US, and other accommodating IOLs have been developed since then. Additionally, considering that our analysis result was to observe macroscale changes, it can be inferred that the AIOL development trend is similar to our results. However, the results in the ClinicalTrials database were different because the AIOL clinical trial was conducted before the request to register data in the ClinicalTrials database; hence, not all the information was included [45].

We can link the time of appearance of AIOLs with the global standard. This is also consistent with the amendments to ISO 11979, the national standard for IOLs. ISO 11979-1, a standard for vocabulary used to evaluate IOLs, was established in 1999, and it has now been revised three times (2006, 2012, and 2018). Among them, the definition of AIOL was included in 2012. Additionally, the AIOL measurement method was included in the dioptric power measurement method standard ISO 11979-2 in 2014. ISO 11979-7, which contains the requirements for IOL clinical investigation, also included a requirement for AIOLs in 2018. Therefore, it can be predicted that AIOL development has been active since 2010; therefore, a global standard for AIOL evaluation is needed. The American National Standards Institute, Inc. (ANSI) in the US also developed a standard for AIOLs in 2015 (ANSI Z80.29). ISO 11979, the global standard for IOLs, and the ANSI for IOLs were first developed in 1999 and 2002, respectively; however, the fact that the standard for AIOLs was developed in 2012–2018 suggests that the AIOL is a relatively recent technology. This is consistent with the results of this study.

In addition to the four major IOL technologies, we found depth of focus and injector-related technologies (Figure 2). Unlike MIOLs, which provide several distinct focal lengths, IOLs with depth-of-focus technology create extended focal lengths [46] and are called extended depth-of-focus (EDOF) IOLs. EDOF IOLs reduce the aberration, glare, and halo that occur in MIOLs without the loss of light while providing vision at various distances through IOL surface processing [46,47]. Therefore, we believe that EDOF IOLs, which correct presbyopia and further develop surface technology, have been recently developed.

Moreover, words related to injectors, such as “cartridges”, “nozzles”, “plungers”, and “tips”, have also been found in patents from the last six years. One of the main goals of recent cataract surgeries is to reduce the postoperative complications and help patients recover quickly [48]. An injector preloaded with an IOL can reduce incorrect IOL insertion, toxic anterior segment syndrome (TASS), and endophthalmitis, which may occur during surgery [49]. When the operators use a syringe-type injector, they can perform the surgery with one hand while fixing the eye with the other hand [48]. By changing the injector design, it is possible to reduce the wound enlargement during IOL insertion. Important design factors are the size of the nozzle, shape of the cushion, and design of the lens cartridge [48]. We believe that the development of injector technology to reduce the side effects during surgery is derived from the word clouds.

So far, the IOL technologies have been derived from the patents, and based on this, we have analyzed how these technologies appear in the papers and clinical trials. First, the patents, papers, and clinical trials were compared for the four major IOL technologies derived through the word clouds and topic modeling (Figure 5). A total of 1615 patents were published by 2021, and the overall number of patents is gradually increasing. It decreased gradually from 1986 to 2000, with no IOL patents from 2000 to 2005; however, from 2006, IOL patents increased rapidly. In the US, the number of recently applied for patents will be higher because patents are opened 18 months after the patent application date. In this study, a total of 169 papers were searched, and although the number of papers published each year is relatively consistent, it can be observed that the number of papers published from 2013 to 2021 was higher than before. Additionally, a total of 454 clinical trials were searched through the ClinicalTrials database, and it was found that many clinical trials have been conducted rapidly since 2010.

Data lanes were derived to analyze the development patterns of the IOL technology shown in the patents, papers, and clinical trials (Figure 6). Since our purpose was to compare the data from the patents, papers, and clinical trials, we only included data from 2005 to 2021, which included all three datasets. The sum of the patents, papers, and clinical trials is the sum from 2005 to 2021, and the circle, square, and triangle sizes are different in the four steps to visualize the amount of each type of data. Additionally, we showed the numbers of patents, papers, and clinical trials in the last five years and the percentage of them in each data from 2005 to 2021. We also underlined the technology that had a high rate of patents, papers, and clinical trials in the last five years. These are surface, haptic, and material, respectively.

For the recent IOL technologies that have appeared in the last six years, the patents, papers, and clinical trials were compared and analyzed for the EDOF- and injector-related technologies (Table 2). Both technologies appear more frequently in the patents than in the papers, showing the characteristics of a company-led technology field. This is because, in general, companies publish more patents than papers compared to universities [20]. In particular, the number of patents for EDOF technologies is increasing rapidly, and considering the unpublished patents in 2021, the number of patents related to EDOF technologies may increase further. Additionally, because the number of clinical trials related to EDOF technologies is increasing, it is expected that more EDOF IOLs will be developed in the future.

Unlike EDOF technologies, the number of patents for injector-related technologies was significantly higher than the number of papers and clinical trials. Several companies have developed various types of injectors to increase the IOL effectiveness. However, clinical trials that prove clinical safety, such as the side effects of using the injector, are limited, and there is a possibility that clinical trials to prove this will increase in the future.

These results were verified using a focus group interview. Five experts supported our research methods and results, and they provided valid explanations to support our findings. The expert in IOL manufacturing suggested that the change in material and surface technology development could be an attempt to reduce the size of the eye incision during cataract surgery to help patients recover faster and reduce side effects, such as the risk of astigmatism. If the IOL material is flexible, it can be folded when being inserted into the eye, reducing the size of the eye incision. With an aspheric surface, it can be made thinner than spherical IOLs. The ophthalmologist reported that in the early stages of IOL development, technology was developed to increase the biocompatibility, such as reducing the inflammatory response. In recent years, multifocal IOLs and accommodation technologies that correct presbyopia so that patients can see all distances without glasses have mainly been developed. Additionally, the IOL tester from a nationally accredited testing institute in the focus groups said that there was a growing demand for IOL testing involving different types of injectors. This is consistent with what we found in the word cloud and topic modeling.

## 4. Discussion

Based on the results so far, we were able to identify the technological development patterns of IOLs over time as follows. First, there had not been as much research and technology development on materials compared to the other three subjects, although clinical trials on materials are still being conducted. This is because, although the development of materials suitable for IOLs is almost complete, clinical trial studies to compare the clinical safety of raw materials are still being conducted. Despite the inactive development of materials with new characteristics, the fact that several clinical trials related to these materials are available suggests that raw materials play an important role in the clinical success of IOLs.

Second, compared to materials, research and technology development and clinical trials of haptic, surface, and accommodation are active. Additionally, from the topic modeling, these three themes were found to be highly related to each other, which can be interpreted as active research and the development of products to correct presbyopia by restoring accommodative power (2011–2015, topics 1 and 5, Figure 7).

Third, it appears that several patents and papers on haptic- and accommodation-related technologies are available. Considering that one of the principles of performing the accommodation function in the eye is to facilitate haptic movement, it is possible to exert accommodative power on the eye through the haptic. It appears that accommodation-related technologies are being actively developed. However, compared to the number of patents and papers, the number of clinical trials is small, meaning there is a high possibility that a clinical trial to verify the clinical effectiveness of accommodation technology will be established in the future.

Lastly, EDOF- and injector-related technologies have emerged recently, and all are being developed by medical device companies. As the effectiveness and safety of IOLs are often proved in clinical trials, clinical trials of these two technologies are likely to increase.

So far, we have derived the major and new IOL technologies based on the patents and compared the technology development and clinical trials. Here, we discuss how regulations manage those two types of IOLs technologies. Many countries evaluate the safety and effectiveness of IOLs according to ISO 11979, the international standard for IOLs. The major four technologies we derived are managed as follows.

First, haptics are parts that fix IOLs to the eye, and they are related to the mechanical properties of IOLs. Axial displacement, optic tilt, and optical displacement tests in ISO 11979-3 [50], which is the specification of IOLs for mechanical properties, are related to haptics [51]. Moreover, ISO 11979-3 includes dynamic fatigue durability and loop pull strength. They evaluate the strength of the haptic so that the IOL can be stably positioned in the eye without breakage [50].

Second, IOL material evaluation is related to biocompatibility (ISO 11979-5) and clinical investigation (ISO 11979-7). Biocompatibility involves testing the safety of materials on animals before applying the IOL to humans. For IOLs, cytotoxicity, sensitization, genotoxicity, local effect after implantation, and ocular implantation tests are required after risk assessment [50]. Additionally, unlike other medical devices, exhaustive extraction, leachable, hydrolytic stability, photostability against UV/Vis irradiation, stability against Nd-YAG laser exposure, and insoluble inorganics are also evaluated for IOLs [50]. ISO 11979-7 guides the characteristics of IOLs that should be evaluated in clinical investigations [50].

The third technology, the surface-related technology, was initially related to biocompatibility, although it was gradually developed in the direction of increasing the independence of the patient’s glasses. The performance of IOLs with two or more focal points can be evaluated through the optical property test in ISO 11979-2 [50]. In this standard, test methods for evaluating products with improved optical properties, such as MIOLs and AIOLs, and the tolerance limits for each IOL are presented [50].

Finally, the method for evaluating the IOL applied accommodation technology was included when ISO 11979-2, 3, and 7 were revised, respectively [50]. ISO 11979-2 suggested a method for evaluating the dioptric power and imaging quality of AIOLs. ISO 11979-3 states that AIOLs need centration and tilt tests, and it should be confirmed that the movement is repeated for at least 1 million cycles and that the movement or shape of the AIOLs should be evaluated in terms of whether it does not change when aging [50]. Moreover, ISO 11979-7 guides additional considerations in clinical trials using AIOLs [50].

Although the major IOL technologies have been evaluated as above, in the case of emerging technologies, the regulations of regulatory authorities have not yet kept pace with the IOL technology. In particular, for IOLs incorporating EDOF technology, it is necessary to evaluate whether they provide the developer’s intended vision at an extended focal length; however, regulatory authorities have not provided an evaluation method for this. Generally, regulatory authorities use an evaluation method based on internationally agreed standards for harmonization. However, the international standard for IOLs does not provide a standardized method for evaluating the performance of EDOF IOLs. In 2018, the American National Standard for EDOF IOLs was established. However, because this is not an international standard, there is no obligation to apply it to any regulatory body other than that in the US. Therefore, regulatory authorities should quickly prepare guidelines for EDOF performance evaluation and review the international standards through the exchange of opinions with other countries. Thus, it is necessary to support the use of EDOF IOLs, which have fewer side effects than existing multifocal IOLs and provide a wider range of vision to patients.

## 5. Conclusions

In this study, we suggest that medical device technology development pattern analysis is necessary to understand healthcare technology, while regulatory authorities can find innovative healthcare products and support them quickly if our method is applied. We developed a method that can be used for product identification and mined data based on text using three types of data sources: patents, papers, and clinical trials.

This study provides three implications. First, healthcare product developers, researchers, and managers should use these methods to obtain the information they need to develop their products. To develop innovative products, information on technology development trends in the field and technology development patterns for similar products is helpful. Therefore, developers and researchers who want to develop new products should use our method, which provides detailed product development steps for decision making during product development.

Second, each regulatory authority needs to utilize our proposed method to rapidly detect innovative products so as to benefit patients and enhance the country’s competitiveness by supporting industries. The role of regulatory authorities is not limited to verifying the safety and effectiveness of healthcare products, although several regulatory authorities do not detect products with innovative technologies. The results of our study proved that it can be applied by regulatory authorities to facilitate the discovery of innovative healthcare products as well as used in the product identification stage.

Finally, regulators should develop guidelines for evaluating products using new technologies in a timely manner. This is because a regulatory authority’s preemptive guidelines support rapid product development and the approval of healthcare product companies. In the case of the IOLs presented as an example, EDOF technology appeared, although each regulatory authority did not provide guidelines to evaluate it objectively and scientifically. In addition to IOLs, many products with new technologies must have been developed.

In addition, medical device developers try to develop medical devices by reflecting the clinical environment of medical devices in order to improve their safety and effectiveness [52,53], while regulatory authorities also need to improve their review methods according to the evolution of development and verification technology. Medical device developers used computation fluid dynamic [52] and patient-oriented data [53] to develop their medical devices, and thus regulators need to use new tools such as in silico, real-world data.

Although this study proposed a new method that can be used to search for innovative healthcare products, it had the following limitations. First, we used three types of data: patents, papers, and clinical trials; however, other data can also be applied depending on the analysis purpose. For example, data obtained from an industry association, media data, and data issued by a regulatory agency may be used. The data used in this study only include information on technology development and clinical trials in the US, without considering those from Europe and Asia. Additionally, since we used US patents provided by the KIPRIS, the data may not completely match the actual US patents. Finally, it takes 18 months for a patent to be published, while it takes a considerable period for a paper to be published. Therefore, the time of comparison of the three datasets may differ from the time when the actual product was researched and developed. Additionally, although we have applied this method to IOLs, which have a long development history and are under active development, it has never been applied to other healthcare products with inconsistent development patterns. Moreover, topic modeling has a limitation in that not all data can be used by cutting off infrequent data for analysis easiness. In addition, we interviewed five experts related to IOLs, but there may be a limit to the number of experts.

Therefore, IOL technology analysis through global data and new information may be needed in the future to improve our study. In addition, verification through other healthcare products, diversification of the analysis methods, and reinforcement of the expert reviews may be required in the future. Moreover, while this study is about the co-evolution of medical device technology and regulation, we believe that the evolution of medical device development and validation technology and the evolution of regulation will provide new insights into technology and regulation. Therefore, it is believed that further research on this will provide meaningful information on the co-evolution of technology and regulation.

## Figures and Tables

**Figure 1 healthcare-11-00453-f001:**
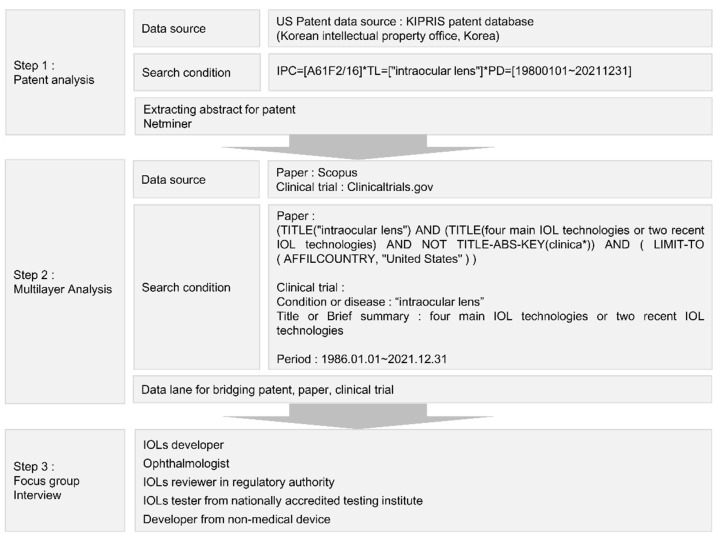
Methodological framework.

**Figure 2 healthcare-11-00453-f002:**
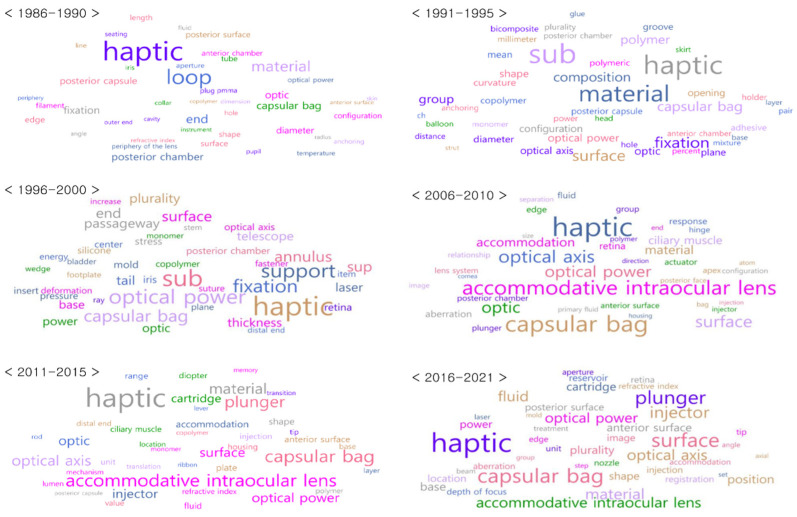
Word cloud of patents for the IOL technology analysis from 1986–2021.

**Figure 3 healthcare-11-00453-f003:**
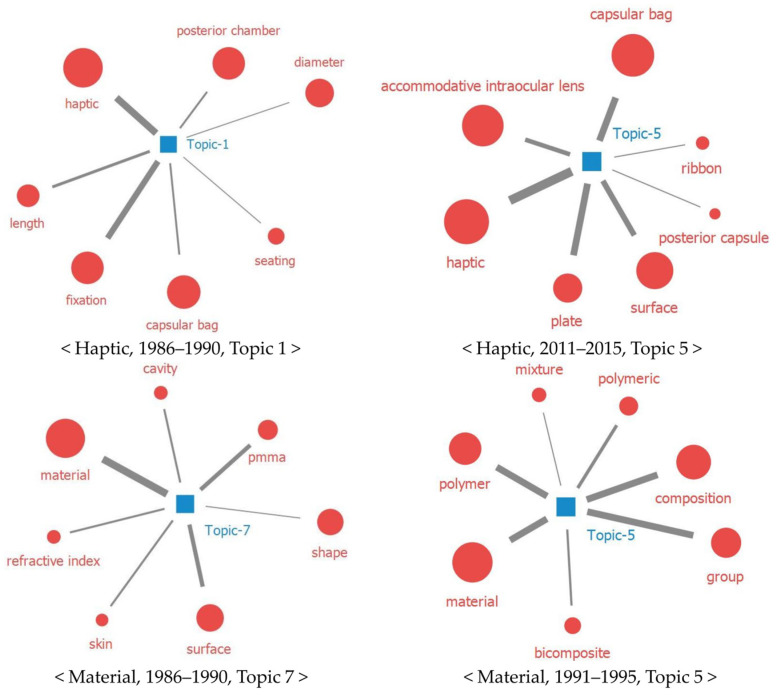
Topic modeling results for haptic and material among the 4 IOL technologies.

**Figure 4 healthcare-11-00453-f004:**
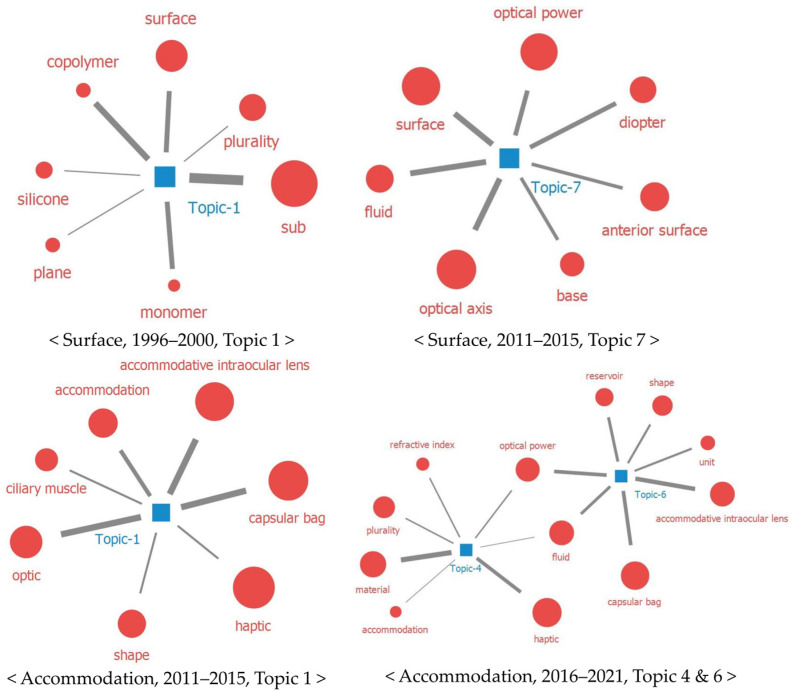
Topic modeling results for surface and accommodation among the 4 IOL technologies.

**Figure 5 healthcare-11-00453-f005:**
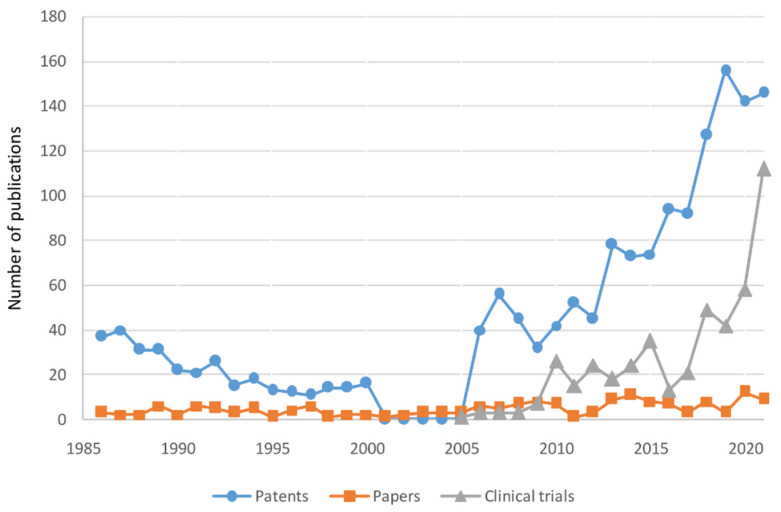
Number of patents, papers, and clinical trials for IOLs by year.

**Figure 6 healthcare-11-00453-f006:**
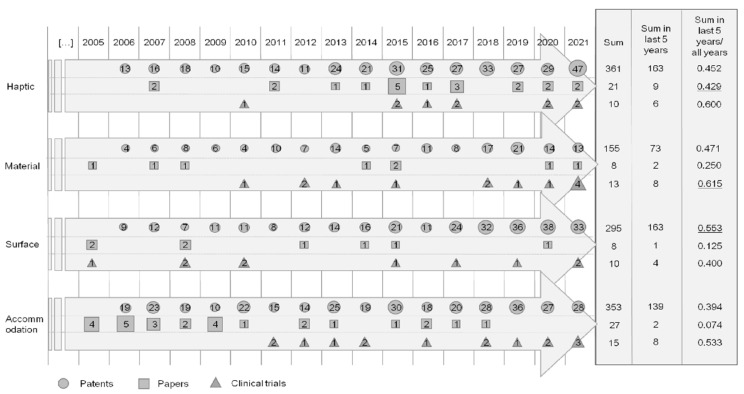
Data lanes for the IOL technology from the patents, papers, and clinical trials.

**Figure 7 healthcare-11-00453-f007:**
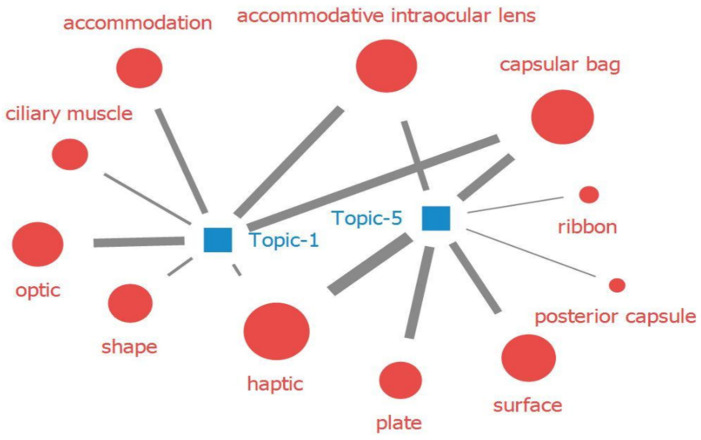
Topic modeling results showing the high correlation among haptic, accommodation, and surface.

**Table 1 healthcare-11-00453-t001:** Focus group members and interview questions.

Focus Group	Interview Questions
- IOL developer - Ophthalmologist - IOL reviewer in a regulatory authority - IOL tester from a nationally accredited testing institute - Developer from non-medical device field	- Validity of the research method - Suitability of IOL technologies derived through the word cloud and topic modeling - Interpretation validity of the relationships of the words in the topic modeling - Validity of the explanations for the changes in technologies over time - Suitability of the explanations for bridging the patents, papers, and clinical trials - Suitability of the conclusions drawn by bridging the patents, papers, and clinical trials - Other expert opinions from the analysis results

**Table 2 healthcare-11-00453-t002:** Number of patents, papers, and clinical trials on recent technologies.

Technology		2016	2017	2018	2019	2020	2021	Sum
**EDOF**	Patents	2	3	6	4	12	2	28
	Papers	-	1	2	-	-	-	3
	Clinical Trials	-	1	2	4	5	5	17
**Injector**	Patents	7	5	13	9	13	13	60
	Papers	-	-	-	2	-	-	2
	Clinical Trials	-	-	-	-	-	1	1

## Data Availability

The data used in this study are available to other authors who require access to this material.

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
