# Peer review of "Development Pattern of Medical Device Technology and Regulatory Evolution of Cataract Treatment"

_healthcare, 2023, doi:10.3390/healthcare11040453_

Round 1

Reviewer 1 Report

The authors of the paper have carried out a study on healthcare technologies and regulations to evaluate this technology by carrying out an example study on the use of IntraOcular lenses for cataract treatment.

The paper is well documented and the reviewer has some minor concerns:

1.       Please modify the figures 2, 3,7 since the grey lines are overshadowing the words. Please place the words above the lines.

2.       Please cite these papers Design of a novel heating device for infusion fluids in vitrectomy (https://doi.org/10.1016/j.applthermaleng.2017.08.027), A novel patient-oriented numerical procedure for glaucoma drainage devices (DOI: 10.1002/cnm.3141)

Author Response

Response to Reviewer 1 Comments

Thank you very much for all your comments. We are very glad that you give us the opportunity to revise our manuscript. We have worked hard to address your excellent and detailed comments. Based on them, we have substantially improved this manuscript. We hope you like the changes we made.

(Comment #1)

The authors of the paper have carried out a study on healthcare technologies and regulations to evaluate this technology by carrying out an example study on the use of IntraOcular lenses for cataract treatment. The paper is well documented and the reviewer has some minor concerns:

  1. Please modify the figures 2, 3,7 since the grey lines are overshadowing the words. Please place the words above the lines.

Response for Comment 1. (in red)

Answer

Thank you for your kind comment. We agreed that some lines were overshadowing the words. We modified the picture so that the lines do not cover the text.

Modification #1 (line 238 in revised manuscript)

Figure 3. Topic modeling results for haptic and material among the 4 technologies of IOLs.

Modification #2 (line 240 in revised manuscript)

Figure 4. Topic modeling results for surface and accommodation among the 4 technologies of IOLs.

Modification #3 (line 434 in revised manuscript)

Figure 7. Topic modeling result showing high correlation with haptic, accommodation, and surface.

(Comment #2)

  1. Please cite these papers Design of a novel heating device for infusion fluids in vitrectomy (https://doi.org/10.1016/j.applthermaleng.2017.08.027), A novel patient-oriented numerical procedure for glaucoma drainage devices (DOI: 10.1002/cnm.3141)

Response for Comment 2. (in red)

Answer

Thank you for giving additional insight to improve our paper. According to your comment, we added to the conclusion that the regulatory authorities need to evolve review methods according to the evolution of medical device development and verification technology (computational fluid dynamic, use of patient-centered data, etc.), and future study on the coevolution between medical device development, verification technology and regulation is also necessary.

Modification #1 (line 521 – line 526 in revised manuscript)

                In addition, medical device developers try to develop medical devices by reflecting the clinical environment of medical devices to improve their safety and effectiveness [52], [53], and regulatory authorities also need to improve their review methods according to the evolution of development and verification technology. Medical device developers used computation fluid dynamic [52] and patient-oriented data [53] to develop their medical device, and thus regulators need to use new tools such as in silico, real world data.

Modification #2 (line 543 – line 551 in revised manuscript)

Therefore, IOLs technology analysis through global data and new information may be needed in the future to improve our study. In addition, verification through other healthcare products, diversification of analysis methods, and reinforcement of expert review may be required in the future. Moreover, while this study is about the co-evolution of medical device technology and regulation, we believe that the evolution of medical device development and validation technology and the evolution of regulation will provide new insights into technology and regulation. Therefore, it is believed that further research on this will provide meaningful information on the co-evolution of technology and regulation.

Reviewer 2 Report

Overall, this is a well-written and interesting paper. Also, I think it is a timely paper written for related researchers and experts who are interested in healthcare technology development patterns. However, there are some parts that are not clear. I have several suggestions for improving this manuscript.

Introduction:

ž   Overall, the background of this study is well-written, but the methodology is mixed in the Introduction section. Please clearly divide the contents of the methodology into how-to sessions.

Methods:

ž   A relatively systematic methodology was used, which was well-written and presented with references.

ž   When you want to grasp technology in patent, why did you start the period from 1980?

ž   In the description of the LDA method, please move to the result section for the description of the optimal number and Figure 1.

ž   Why are the search periods in line 125 and line 165 different?

ž   The implementation of FGI and its composition is good. However, what was the subject of the interview (including the list of questions)?

Results: & Discussion

ž   Keep results and discussion as separate as possible so readers can understand your research more clearly.

ž   Figure 4 needs to move to the Method section. Figure 4 is missing about ‘US patents.’

ž   What do the underlined numbers in Figure 6 mean?

ž   When presenting the results and discussion of FGI, readers will be able to get more information if you present whether the experts are clinical experts or regulatory experts rather than naming just ‘one expert’.

Author Response

Response to Reviewer 2 Comments

Thank you very much for all your comments. We are very glad that you give us the opportunity to revise our manuscript. We have worked hard to address your excellent and detailed comments. Based on them, we have substantially improved this manuscript. We hope you like the changes we made.

(Comment #1)

Overall, this is a well-written and interesting paper. Also, I think it is a timely paper written for related researchers and experts who are interested in healthcare technology development patterns. However, there are some parts that are not clear. I have several suggestions for improving this manuscript.

Introduction: Overall, the background of this study is well-written, but the methodology is mixed in the Introduction section. Please clearly divide the contents of the methodology into how-to sessions.

Response for Comment 1. (in red)

Answer

We sincerely thank you for pushing us to improve the introduction section. We identified some methodologies intermingled in the section. So we deleted some methodology in the section based on your comment. It could facilitate the logical flow of this study.

Modification #1 (line 53 – line 79 in revised manuscript)

Although analysis of medical device development pattern is necessary to understand healthcare technology, there are few studies on medical device development pattern. Moreover, there are few studies on development pattern including clinical trials, which are necessary for verification of effectiveness and safety of medical devices. In other fields, like chemistry and biotechnology, there are many studies analyzing of new technologies by comparing patents and literature [10]. However, research on medical device technology development patterns was conducted only on wearable devices and smart health monitoring systems [11,12]. They are relatively recent technologies and have limitations in generalizing them to medical devices. Additionally, an important consideration in the medical device development pattern research is that medical devices must be approved by the regulatory authorities. However, there have been few studies that analyzed technology development patterns including clinical trial data required for medical device approval. Therefore, in this study, we proposed a new multilayer analysis method with patents, papers, and clinical trials to analyze medical device development pattern to understand healthcare technologies.

This study will contribute to coevolution of medical device technology and regulation. Current regulatory authorities are trying to enhance national competitiveness by not only approving healthcare products, but also detecting and supporting innovative healthcare products. This study indicated that even regulatory authorities can find innovative healthcare products based on objective data. It can also be used for regulation reviews, and guideline development for safety and effectiveness evaluation. Furthermore, multilayer analysis using patents, papers, and clinical trial data can help the government, developers, and managers in decision-making by providing information on technology development stages, such as research in academia, company-led development, and clinical trials [20]. And this result will be able to provide useful information for the healthcare industry by increasing the understanding of the technology development pattern of healthcare products.

(Comment#2)

Methods: ž   A relatively systematic methodology was used, which was well-written and presented with references.

Response for Comment 2. (in red)

Answer

Thank you for your kind comment. This study used a relatively systematic methodology based on the confirmed previous studies. We, all authors appreciate your valuable comment.

(Comment#3)

Methods: ž   When you want to grasp technology in patent, why did you start the period from 1980?

Response for Comment 3. (in red)

Answer

Thank you for your comment. We tried to collect as much patent data as possible from Korea Intellectual Property Rights Information Service (KIPRIS) database, which was provided by Korean Intellectual Property Office (KIPO). In fact, they provide that database in cooperation with the US Patent and Trademark Office (USPTO). However, we couldn’t obtain data before 1980 from them. For clarification, we added that explanation in methodology section. We disclosed that we used US patent data provided by the KIPO, in the limitation section.

Modification #1 (line 112 – line 122 in revised manuscript)

We searched for US patents among the patents provided by Korean Intellectual Property Office (KIPO), using the Korea Intellectual Property Rights Information Service (KIPRIS) of the KIPO. We used US patents because the US is the largest of the global IOLs market and three of the five companies (Alcon, Inc., Johnson and Johnson Vision Care, and Bausch + Lomb) leading the global market are US companies [27]. We used A61F2/16, an International Patent Classification corresponding to the IOL, as a search term. However, because patents on the IOL apparatus or IOL insertion method were searched, we additionally selected patents with “intraocular lens” in the title to reduce false positives. The search period was from 1980 to 2021. It is because although we tried to collect as much data as possible on IOLs, we could not obtain patents for IOLs prior to 1980 through KIPRIS.

Modification #2 (line 154 – line 164 in revised manuscript)

In the last stage of patent analysis, we analyzed the frequency of the appearance of refined keywords and visualized them through a word cloud. In this way, we were able to derive the main topics of IOL technology and topic words. Additionally, to derive recent technology, it is necessary to observe the change in technology over time; therefore, we analyzed patents by dividing them into five-year units. Although we set search period from 1980 to 2021, but since the US IOL patents matching our searching conditions were searched since 1986, we analyzed them by dividing them into time periods of 1986–1990, 1991–1995, 1996–2000, 2001–2005, 2006–2010, 2011–2015, and 2016–2021. However, we were unable to obtain the analysis results for 2001–2005 because no IOL-related patents were searched during this period. And patent data published in 2021 were combined with data from 2016–2020.

Modification #3 (line 527 – line 542 in revised manuscript)

Although this study proposed a new method that can be used to search for innovative healthcare products, it has the following limitations. First, we used three types of data: patents, papers, and clinical trials; however, other data can also be applied depending on the analysis purpose. For example, data obtained from an industry association, media data, and data issued by a regulatory agency may be used. The data used in this study only includes information on technology development and clinical trials in the US without considering those from Europe and Asia. Additionally, since we used US patents provided by KIPRIS, it may not completely match the actual US patent. Finally, it takes 18 months for a patent to be published, and a considerable period for a paper to be published. Therefore, the time of comparison of the three datasets may differ from the time when the actual product is researched and developed. Additionally, although we have applied this method with IOLs that have a long development history and are under active development, it has never been applied to other healthcare products with inconsistent development pattern. Moreover, topic modeling has a limitation in that not all data can be used by cutting off infrequent data for analysis easiness. Also, we interviewed five experts related to IOLs, there may be a limit to the number of experts.

(Comment#4)

Methods: ž   In the description of the LDA method, please move to the result section for the description of the optimal number and Figure 1.

Response for Comment 4. (in red)

Answer

Thank you for your kind comment. We agreed that it confused the readers that the results such as description of the optimal number and figure 1 were in those positions. So this study moved them to the result section as you commented. At the same time we added some more descriptions for LDA method.

Modification #1 (line 147 – line 153 in revised manuscript)

The third step involved the derivation of topics related to IOLs shown in the patents and keywords constituting each topic through topic modeling. The Latent Dirichlet Allocation (LDA) algorithm was used in topic modeling. LDA is a probabilistic modeling method that finds topics through words in each patent and determines the words that are placed in the topic. Since LDA is an unsupervised generative model without prior information about topics and words, we can easily find all words related to the topic. However, it is not suitable for a small amount of data of a normally distributed environment [29].

Modification #2 (line 242 – line 246 in revised manuscript)

When performing topic modeling, the number of topics to be extracted must be specified. In previous studies, analysis was performed by designating 5–10 topics; the number of topics was determined based on interpretability after analysis by repeatedly changing the number of topics [32]. This study was analyzed in the same way, and the optimal number of topics was 7.

(Comment#5)

Methods: ž   Why are the search periods in line 125 and line 165 different?

Response for Comment 5. (in red)

Answer

Thank you for this comment. We tried to collect as much patent data as possible from KIPRIS, but we couldn’t obtain data before 1980, and thus we defined the search period from 1980 to 2021. However, in reality, we could obtain patents from 1986 according to our search conditions (patent classification, including intraocular lens in the patent title). The period from 1980 to 2021 is the entire search period in line 125 of the previous version. It was 1986, which had been described in line 165, when the results began to appear during the search period. So, the period of line 125 and line 165 was different. For clarification, we added more explanation in the methodology section.

Modification #1 (line 112 – line 122 in revised manuscript)

We searched for US patents among the patents provided Korean Intellectual Property Office (KIPO), using the Korea Intellectual Property Rights Information Service (KIPRIS) of the KIPO. We used US patents because the US is the largest of the global IOLs market and three of the five companies (Alcon, Inc., Johnson and Johnson Vision Care, and Bausch + Lomb) leading the global market are US companies [27]. We used A61F2/16, an International Patent Classification corresponding to the IOL, as a search term. However, because patents on the IOL apparatus or IOL insertion method were searched, we additionally selected patents with “intraocular lens” in the title to reduce false positives. The search period was from 1980 to 2021. It is because although we tried to collect as much data as possible on IOLs, we could not obtain patents for IOLs prior to 1980 through KIPRIS.

Modification #2 (line 154 – line 164 in revised manuscript)

In the last stage of patent analysis, we analyzed the frequency of the appearance of refined keywords and visualized them through a word cloud. In this way, we were able to derive the main topics of IOL technology and topic words. Additionally, to derive recent technology, it is necessary to observe the change in technology over time; therefore, we analyzed patents by dividing them into five-year units. Although we set search period from 1980 to 2021, but since the US IOL patents matching our searching conditions were searched since 1986, we analyzed them by dividing them into time periods of 1986–1990, 1991–1995, 1996–2000, 2001–2005, 2006–2010, 2011–2015, and 2016–2021. However, we were unable to obtain the analysis results for 2001–2005 because no IOL-related patents were searched during this period. And patent data published in 2021 were combined with data from 2016–2020.

Modification #3 (line 527 – line 542 in revised manuscript)

Although this study proposed a new method that can be used to search for innovative healthcare products, it has the following limitations. First, we used three types of data: patents, papers, and clinical trials; however, other data can also be applied depending on the analysis purpose. For example, data obtained from an industry association, media data, and data issued by a regulatory agency may be used. The data used in this study only includes information on technology development and clinical trials in the US without considering those from Europe and Asia. Additionally, since we used US patents provided by KIPRIS, it may not completely match the actual US patent. Finally, it takes 18 months for a patent to be published, and a considerable period for a paper to be published. Therefore, the time of comparison of the three datasets may differ from the time when the actual product is researched and developed. Additionally, although we have applied this method with IOLs that have a long development history and are under active development, it has never been applied to other healthcare products with inconsistent development pattern. Moreover, topic modeling has a limitation in that not all data can be used by cutting off infrequent data for analysis easiness. Also, we interviewed five experts related to IOLs, there may be a limit to the number of experts.

(Comment#6)

Methods: ž   The implementation of FGI and its composition is good. However, what was the subject of the interview (including the list  of questions)?

Response for Comment 6. (in red)

Answer

Thank you for your kind comment. According to your comment, we added more detailed information about focus group interview. Readers can find out who the experts are and what questions they were asked. The information are summarized in Table 1.

Modification #1 (line 201 – line 211 in revised manuscript)

We selected a focus group of experts and verified the results of our analysis. We wanted to verify the validity of our analysis method and results, and for this we conducted focus group interviews. Five experts were selected for the study. First, we included an IOL developer because we wanted to include experts in IOL R&D. The expert was also in charge of developing national standards for the field of ophthalmology in Korea. Additionally, we included clinicians who selected IOLs and performed cataract surgery in the focus group. We also included a reviewer from regulatory authorities in the Korean government and an expert who tested the performance of IOLs in the nationally accredited testing institute on the focus group. Finally, to prevent the focus group from being composed of only medical experts, we verified our analysis results by including experts with extensive experience in technology development in the non-medical device field.

Modification #2 (line 212 – line 226 in revised manuscript)

Based on our results, we prepared interview questions in advance before meeting with each expert. The questions included research methods and research results, and each question is as follows.

  • Validity of research method
  • Suitability of IOLs technologies derived through word cloud and topic modeling
  • Interpretation validity of relationship of words in topic modeling
  • Validity of explanations for changes in technologies over time
  • Suitability of the explanations of bridging of patents, papers, and clinical trials
  • Suitability of conclusions drawn by bridging patent, papers, and clinical trials
  • Other expert opinions from the analysis result

We asked the same questions to five experts and our questions are listed in Table 1.

Table 1. Focus group member and interview questions

 Focus Group

Interview questions

-      IOLs developer

-      Ophthalmologist

-      IOLs reviewer in regulatory authority

-      IOLs tester from nationally accredited testing institute

-      Developer from non-medical device field

-      Validity of research method

-      Suitability of IOLs technologies derived through word cloud and topic modeling

-      Interpretation validity of relationship of words in topic modeling

-      Validity of explanations for changes in technologies over time

-      Suitability of the explanations of bridging of patents, papers, and clinical trials

-      Suitability of conclusions drawn by bridging patent, papers, and clinical trials

-      Other expert opinions from the analysis result

(Comment#7)

Results: & Discussion: ž   Keep results and discussion as separate as possible so readers can understand your research more clearly.

Response for Comment 7 (in red)

Answer

Thank you for your kind comment. According to your comment, we divided results and discussion. In the results section, we described the word cloud and topic modeling analysis results of patents, patents-papers-clinical trials bridging analysis, and focus group interviews. And in the discussion section, the results were integrated to describe the findings on the development pattern of intraocular lens technology and how the current regulatory authorities are dealing with medical device technology.

Modification #1 (line 228 in revised manuscript)

  1. Results

Modification #2 (line 242 - line 246 in revised manuscript)

When performing topic modeling, the number of topics to be extracted must be specified. In previous studies, analysis was performed by designating 5–10 topics; the number of topics was determined based on interpretability after analysis by repeatedly changing the number of topics [32]. This study was analyzed in the same way, and the optimal number of topics was 7.

Modification #3 (line 247 - line 254 in revised manuscript)

The following can be derived for each of the four major IOLs technology from topic modeling. First, haptic, which is closely related to IOL stability, has been developed in various forms to increase IOL stability, and has recently become the accommodation core of AIOLs. From topic modeling, haptic was highly correlated with the posterior chamber, capsular bag, and fixation (1986–1990, topic 1). As haptic is highly related to IOL stability, it is considered to be highly related to the success or failure of IOL implantation surgery [33]. Particularly, haptic is known to be highly related to the intraocular stability of products with enhanced optical functions, such as multifocal (MIOLs) and toric IOLs [33].

Modification #4 (line 376 - line 384 in revised manuscript)

Data lanes were derived to analyze the development patterns of IOL technology shown in patents, papers, and clinical trials (Figure 6). Since our purpose was to compare data from patents, papers, and clinical trials, we only included data from 2005 to 2021, which included all three datasets. The sum of patents, papers, and clinical trials is the sum from 2005 to 2021, and the circle, square, and triangle sizes are different in the four steps to visualize the amount of each data. Additionally, we showed the numbers of patents, papers, and clinical trials in the last 5 years and the percentage of them in each data from 2005 to 2021. We also underlined the technology that had a high rate of patents, papers, and clinical trials in the last 5 years. They are surface, haptic, and material, respectively.

Modification #5 (line 387 - line 417 in revised manuscript)

For the recent IOL technologies which have appeared in the last six years, patents, papers, and clinical trials were compared and analyzed for EDOF- and injector-related technologies. (Table 1) Both technologies appear more frequently in patents than in papers, showing the characteristics of a company-led technology field. This is because, in general, companies publish more patents than papers compared to universities [20]. Particularly, the number of patents for EDOF is increasing rapidly, and considering the unpublished patents in 2021, the number of patents related to EDOF may increase further. Additionally, because the number of clinical trials related to EDOF is increasing, it is expected that more EDOF IOLs will be developed in the future.

Table 1. Number of patents, papers, and clinical trials on recent technologies

Technology

2016

2017

2018

2019

2020

2021

Sum

EDOF

Patents

2

3

6

4

12

2

28

Papers

-

1

2

-

-

-

3

Clinical

Trials

-

1

2

4

5

5

17

Injector

Patents

7

5

13

9

13

13

60

Papers

-

-

-

2

-

-

2

Clinical

Trials

-

-

-

-

-

1

1

Unlike EDOF, the number of patents for injector-related technology was significantly higher than that of papers and clinical trials. Several companies have developed various types of injectors to increase IOL effectiveness. However, clinical trials that prove clinical safety, such as the side effects of using the injector, are limited and there is a possibility that clinical trials to prove this will increase in the future.

These results were verified using a focus group interview. Five experts supported our research methods and results and provided valid explanations to support our findings. The expert in IOLs manufacturer suggested that the change in material and surface technology development could be an attempt to reduce the size of the eye incision during cataract surgery to help patients recover faster and reduce side effects, such as the risk of astigmatism. If the IOL material is flexible, it can be folded when inserting into the eye, reducing the size of the eye incision; with aspheric surface, it can be made thinner than spherical IOLs. The ophthalmologist reported that in the early stages of IOL development, technology was developed to increase biocompatibility, such as reducing the inflammatory response. In recent years, multifocal IOLs and accommodation technologies that correct presbyopia so that patients can see all distances without glasses have been mainly developed. Additionally, the IOLs tester from nationally accredited testing institute in the focus groups said there was a growing demand for IOL testing involving different types of injectors. This is consistent with what we find in word cloud and topic modeling.

Modification #6 (line 419 - line 428 in revised manuscript)

  1. Discussion

Based on the results so far, we were able to identify the technological development pattern of IOLs over time as follows. First, there have not been as much research and technology development on materials compared to the other three subjects, but clinical trials on materials are still being conducted. This is because although the development of materials suitable for IOLs is almost complete, clinical trial studies to compare the clinical safety of raw materials are still being conducted. Despite the inactive development of materials with new characteristics, the fact that several clinical trials related to these materials are available suggests that raw materials play an important role in the clinical success of IOLs.

Modification #7 (line 436 - line 446 in revised manuscript)

Thirdly, it appears that several patents and papers on haptic- and accommodation-related technologies are available. Considering that one of the principles of performing the accommodation function in the eye is to facilitate haptic movement, it is possible to exert accommodative power on the eye through the haptic. It appears that accommodation-related technologies are being actively developed. However, compared to the number of patents and papers, the number of clinical trials is small, so there is a high possibility that a clinical trial to verify the clinical effectiveness of accommodation technology will be established in the future.

Lastly, EDOF and injector-related technologies have emerged recently, and all are being developed by medical device companies. As the effectiveness and safety of IOLs are often proved in clinical trials, clinical trials of these two technologies are likely to increase.

(Comment#8)

Results: & Discussion: ž   Figure 4 needs to move to the Method section. Figure 4 is missing about ‘US patents.’

Response for Comment 8 (in red)

Answer

Thank you for your esteemed comment. Following your recommendation, we moved to Figure 4 to methodology section. This study was able to make the methodology section clearer by clustering scattered methodology. Moreover, we indicated that the data sources are ‘US patents’ in the figure.

Modification #1 (line 90 – line 102 in revised manuscript)

  1. Methods

We evaluated the technological development patterns and newly emerged technologies of IOLs using a text-based analysis method with three types of data: patents, papers, and clinical trials. We used the methodologies of Niemann et al. (2017) [14], Block et al. (2021) [15], and Wustmans et al. (2021) [16] to compare the patents, papers, and clinical trials. In other words, if the words used in each document are similar, it is assumed that the content is also similar. Using this method, Niemann et al. analyzed the similarity between patents, Block et al. analyzed the similarity between patents and papers, and Wustmans et al. analyzed the similarity between patents and trend data. Our research method is illustrated in Figure 1.

Figure 1. Methodological framework.

(Comment#9)

Results: & Discussion: ž   What do the underlined numbers in Figure 6 mean?

Response for Comment 9 (in red)

Answer

Thank you for your comment. We sincerely apologize this confusion in the previous version of the paper. As you commented, this study added the meaning of underlining in the figure 6. This modification provides the clarification of this study.

Modification #1 (line 376 – line 384 in revised manuscript)

Data lanes were derived to analyze the development patterns of IOL technology shown in patents, papers, and clinical trials (Figure 6). Since our purpose was to compare data from patents, papers, and clinical trials, we only included data from 2005 to 2021, which included all three datasets. The sum of patents, papers, and clinical trials is the sum from 2005 to 2021, and the circle, square, and triangle sizes are different in the four steps to visualize the amount of each data. Additionally, we showed the numbers of patents, papers, and clinical trials in the last 5 years and the percentage of them in each data from 2005 to 2021. We also underlined the technology that had a high rate of patents, papers, and clinical trials in the last 5 years. They are surface, haptic, and material, respectively.

(Comment#10)

Results: & Discussion: ž   When presenting the results and discussion of FGI, readers will be able to get more information if you present whether the experts are clinical experts or regulatory experts rather than naming just ‘one expert’.

Response for Comment 10 (in red)

Answer

Thank you for your kind comment. According to your comment, we added the detailed information of experts who attended the focus group interview, and their suggestions. The experts are as followings; the expert in IOLs, the ophthalmologist and the IOLs tester. This modification strengthened the discussion section.

Modification #1 (line 404 – line 417 in revised manuscript)

These results were verified using a focus group interview. Five experts supported our research methods and results and provided valid explanations to support our findings. The expert in IOLs manufacturer suggested that the change in material and surface technology development could be an attempt to reduce the size of the eye incision during cataract surgery to help patients recover faster and reduce side effects, such as the risk of astigmatism. If the IOL material is flexible, it can be folded when inserting into the eye, reducing the size of the eye incision; with aspheric surface, it can be made thinner than spherical IOLs. The ophthalmologist reported that in the early stages of IOL development, technology was developed to increase biocompatibility, such as reducing the inflammatory response. In recent years, multifocal IOLs and accommodation technologies that correct presbyopia so that patients can see all distances without glasses have been mainly developed. Additionally, the IOLs tester from nationally accredited testing institute in the focus groups said there was a growing demand for IOL testing involving different types of injectors. This is consistent with what we find in word cloud and topic modeling.
